# Silodosin Improves Pain and Urinary Frequency in Bladder Pain Syndrome/Interstitial Cystitis Patients

**DOI:** 10.3390/jcm11195659

**Published:** 2022-09-26

**Authors:** Pedro Abreu-Mendes, Beatriz Araújo-Silva, Ana Charrua, Francisco Cruz, Rui Pinto

**Affiliations:** 1Urology Department, Centro Hospitalar e Universitário de São João, 4200-319 Porto, Portugal; 2Faculty of Medicine, University of Porto, 4200-319 Porto, Portugal; 3Translational Neurourology Group, I3 Instituto de Investigação e Inovação em Saúde, University of Porto, 4200-319 Porto, Portugal; 4Biomedicine Department, Faculty of Medicine, University of Porto, 4200-319 Porto, Portugal

**Keywords:** bladder pain syndrome, interstitial cystitis, stress, alpha-blocker, silodosin

## Abstract

Purpose: Bladder Pain Syndrome/Interstitial Cystitis (BPS/IC) is a bladder-related chronic inflammatory disease. Data indicate that stress enhances the excitability of bladder nociceptors through the stimulation of alpha1A-adrenoceptors. Stress is known to play a crucial role in BPS/IC patients. We aimed to assess the efficacy and safety of daily silodosin in refractory BPS/IC female patients and its correlation with stress coping. Materials and Methods: An open-label trial was conducted with 20 refractory BPS/IC patients. Evaluations occurred at baseline and the 8th and 12th weeks. Primary endpoint was bladder pain evaluated by visual analogue scale (VAS). Secondary endpoints included daily frequency, nocturia and maximum voided volume obtained from a 3-day bladder diary, the O’Leary–Sant Symptom Score, and two questions accessing stress coping. Patients initiated daily doses of 8 mg silodosin, which could be titrated to 16 mg. Median values with percentiles 25 and 75 (25; 75) were used. Wilcoxon signed-rank test was used for comparisons. A minimally important difference of 3 points for pain was established to define clinically relevant improvement. Results: Median age was 56 years. Median pain score decreased from 8.00 (6.00; 8.00) at baseline to 4.00 (2.00; 5.50) (*p* < 0.001), meaning that the primary endpoint was reached. Total urinary frequency decreased from 14.00 (13.00; 21.00) to 9.00 (7.50; 11.00) (*p* < 0.05), and all the other secondary endpoints also showed a statistically significant improvement. Eleven patients improved by ≥3 pain points in VAS, meaning that 65% of patients that ended the study protocol achieved clinical significant improvement or, in the full analysis set, that 55% of the 20 initial patients improved significantly. Fourteen (82%) decreased by ≥2 micturitions/day. Overall, the cohort’s stress coping was low. Conclusions: Silodosin can be an effective and well-tolerated treatment for refractory BPS/IC female patients.

## 1. Introduction

Primary Bladder Pain Syndrome/Interstitial Cystitis (BPS/IC) is a chronic disease. Suprapubic pain associated with bladder filling is the cornerstone symptom and is associated with urinary symptoms such as frequency, nocturia and persistent urge to prevent/relieve pain [1]. Effective curative treatment for BPS/IC is yet unknown. This unfortunate situation is due to the poor understanding of BPS/IC etiology. Chronic bacterial infections, the defective glycosaminoglycan layer of the bladder urothelium, the inappropriate activation of mast cells, autoimmune-mediated mechanisms, and autonomic nervous system dysfunction have been suggested [2]. As the modulation of the activity of bladder nociceptors involves complex mechanisms, different partners may participate in their activation [3]. Furthermore, systemic pain syndromes and somatic disorders such as irritable bowel syndrome, chronic fatigue syndrome, and fibromyalgia frequently coexist with BPS/IC [4,5]. This observation has led, in recent years, to another pathophysiological paradigm in which BPS/IC is seen as a local manifestation of a systemic pain condition [2].

Increasing evidence suggests that stress plays a crucial role in BPS/IC [5,6]. Symptoms are aggravated by stressful periods [7]. Patients who have suffered early-in-life traumatic events show a distinctive symptomatic phenotype, with intense pain and a high urinary frequency [8,9]. Pain triggered or exacerbated by stress is thought to reflect an increase in NE release, particularly in BPS/IC patients [10]. The excessive stimulation of α1_A_-Ars enhances the responses of nociceptors, such as transient receptor potential cation channel subfamily V member 1 (TRPV1) [9], resulting in visceral hyperalgesia [10,11,12]. Interestingly, early-in-life insult causes erratic responses to stress later in life, with an increased release of corticotrophin-releasing factor, adrenocorticotrophic hormone, and norepinephrine (NE) [13,14]. Moreover, an excess of NE may leave an imprint on pain pathways during development, with the dysregulation of the hypothalamic–pituitary–adrenal axis [13].

In a recent pre-clinical experiment, adult female mice developed an equivalent of a BPS/IC phenotype after being subjected to an early-life water avoidance stress test (WAS). In this model, the lower abdominal mechanical pain threshold was decreased and the frequency of bladder voiding contractions during filling cystometries was increased, while urinary NE levels were multiplied by a factor of 20. Interestingly, the blockade of α1_A_-ARs by silodosin, although not altering NE levels, normalized the abdominal pain threshold and bladder function of rats subjected to WAS [11].

Silodosin has a safe profile, conferred by its high selectivity for α1_A_-ARs, limiting cardiovascular events [12]. With this in mind, we performed this single-arm study to evaluate the efficacy and tolerability of a daily dose of silodosin in 20 women with BPS/IC refractory to conventional treatments. Simultaneously, we assessed the capacity of stress coping through a brief personality questionnaire.

## 2. Material and Methods

### 2.1. Patients

This study was approved by the local ethics committee (Comissão de Ética do Centro Hospitalar e Universitário de São João) under protocol number 338-21: “Efeito da Silodosina nos doentes com Síndrome Doloroso vesical aos 3 meses”. The trial occurred in a tertiary university hospital with expertise in BPS/IC. Oral and written consent was obtained from patients after explaining the off-label drug use. Patients were informed that silodosin is licensed for the treatment of male LUTS. The rationale for this use was understood. We also stated that the side effects of this drug were extensively known.

Previously, a small cohort of patients outside the study, refractory to every other previous therapy (oral, systemic and intravesical), showed a good response to silodosin.

Twenty women with BPS/IC refractory to previous therapies were enrolled between 2019 and 2020. As a pilot study, without any available data to predict the potential efficacy, power calculations were not carried out.

During the previous follow-up, patients had been advised to adjust their diet, increase physical activity, avoid stressful events, and stop smoking. Patients under any oral treatment in the last two months, other than simple oral analgesia (such as acetaminophen or ibuprofen), were excluded. Patients who received intravesical treatments in the previous six months were also excluded.

### 2.2. Study Design

We designed a 12-week single-arm exploratory study to evaluate the efficacy of silodosin against BPS/IC symptoms.

Patients were evaluated at baseline, at week 8 (8w), and at week 12. The primary outcome was the pain intensity at 12w, assessed by a 0–10 VAS score (in which 10 was the worst pain ever). The secondary outcomes included the daily urinary frequency, the nocturia, and the MVV recorded in a 3-day bladder diary; the OSS problems and symptoms index (0 to 34, being 34 the worst); and the QoL evaluated with the question used in the IPSS questionnaire (from 0 to 6, with six being the worst quality of life). For the definition of a relevant clinical improvement, we considered a decrease in 3 points in the VAS score as the minimally important difference (MID). Decreases of 2 micturitions in total daily urinary frequency and 1 micturition in nocturia were defined as the MID to consider a relevant improvement in these parameters.

Two questions were addressed to the patients at the beginning of the study in order to assess their capacity to cope with stress: the first question was answered yes/no, “*Does stress aggravates or initiate BPS/IC symptoms?*”, and a second question, “*I have a relaxed personality, coping well with stress*“, could be answered on a scale from 1 to 5, with 1 being “I totally disagree with it” and 5 being “I totally agree with it”. After consenting to enter the study, patients began with an 8 mg daily oral dose of silodosin. This dose could be titrated to 16 mg at 8w. Adverse events were assessed and recorded.

### 2.3. Statistical Analysis

Data analysis was performed with the statistical package for social sciences (SPSS version 28.0). Data are presented as medians, with the 25 and 75 percentiles (25; 75). Differences assessed by a non-parametric Wilcoxon signed-rank test (*p* < 0.05) were considered statistically significant. The analysis was carried out *per*-*protocol*, comparing the data from the baseline with that from 12w in patients completing all 12 weeks. The primary end-point was also evaluated, taking into account the dropouts. Patients who left the study before the 12w were excluded from the final analysis but were included for safety exploration.

## 3. Results

### 3.1. Demographics

Patients had a median age of 56 (46; 66) years old. Three patients presented Hunner lesions at cystoscopy. Fifteen of them had previously received at least one treatment of intra-trigonal injection of onabotulinum toxin A six or more months before. Dose escalation to silodosin 16 mg was requested by four patients at 8w.

A total of three patients abandoned the study. Two patients abandoned therapy at week 8 due to lack of effect, without trying dose escalation despite the clinician’s effort to try medication until the 12th week. The third patient was excluded due to the diagnosis of a UTI during the study period, which caused a symptomatic flare up. Two patients failed to attend the 8w visit due to the COVID-19 pandemic but were observed at 12w.

### 3.2. Outcomes

The characteristics of the population at baseline are shown in Table 1. The median value of pain intensity ascertained by VAS was 8.0 (6.0; 8.0); total urinary frequency and nocturia were 14.0 (13.0; 21.0) and 4.0 (2.0; 4.8), respectively. The MVV was 175 mL (150; 250); OSS was 26.0 (24.0; 28.0); and QoL was 5.0 (4.3; 6.0). Concerning the primary endpoint, pain score at 12w, the median VAS score from the 17 patients available was 4.0 (2.0; 5.5), which indicates a significant reduction when compared to the baseline value of 8.0 (6.0; 8.0) (non-parametric Wilcoxon signed-rank test *p* = 0.001). A total of 11 out of 17 patients (65%) had an improvement of 3 or more points from baseline to 12w, as seen in Figure 1. This reflects an MID improvement in 11 out of 17 patients (65%) at 12w, in the *per-protocol* analysis. When evaluated taking into account the full analysis set, the percentage of clinically significant improvement was 55% (11/20).

At 12w, the total urinary frequency decreased to a median of 9.0 (7.5; 11.0), which was significantly lower than the baseline median of 14.0 (13.0; 21.0) micturition (non-parametric Wilcoxon signed-rank test *p* = 0.01). A total of 14 patients out of the 17 (82%) achieved the MID of a decrease of at least two micturition from baseline to 12w. The total urinary frequency variation per patient result is shown in Figure 2.

Nocturia decreased to 2.0 (1.0; 3.0) at 12w, with the baseline being 4.0 (2.0; 5.5) micturition per night, and this is a statistically significant difference (non-parametric Wilcoxon signed-rank test *p* = 0.01). A total of 13 (76%) patients achieved an MID of one episode per night between baseline and 12w.

Assuming that greater satisfaction relies upon an important improvement in both pain and urinary frequency, we analyzed the simultaneous decrease in pain and total frequency according to the defined MID. A total of 10 out of 17 patients (59%) presented this simultaneous improvement, as shown in Figure 3. Five patients achieved an MID improvement in only one of the parameters and only 2 patients did not reach an MID improvement in any of the two outcomes.

When it comes to the other secondary endpoints, at 12w the median MVV increased to 250 mL (200; 350) mL; OSS total score decreased to 19.00 (14.00; 22.50) with a and QoL decreased to 3.00 (2.00; 4.0). All these data were statistically significant when compared to baseline values through the non-parametric Wilcoxon signed-rank test (MVV *p* = 0.03; OSS *p* = 0.01 and QoL *p* = 0.04) shown in Table 1.

At 8w, some improvement could already be detected. The median VAS pain score was 5.0 (3,0; 6,0), indicating a significantly early pain reduction (non-parametric Wilcoxon signed-rank test *p* = 0.001). Moreover, at this earlier time point, the total frequency had already significantly decreased to 10.0 (8.0; 14.0) micturition when compared to baseline (non-parametric Wilcoxon signed-rank test *p* = 0.03) (Figure 1 and Figure 2).

In an exploratory attempt to find possible predictors of a positive response to silodosin, we compared the demographics between patients who presented an improvement of more than 3 points in the VAS pain intensity and the patients who did not reach this improvement. The compared demographics were: age, pain intensity, urinary frequency, nocturia, MVV, OSS, and QoL. The results and the comparison between groups are presented in Table 2. None of the above characteristics show a statistically significant difference.

Of the 17 patients, 13 answered to the question “Does stress aggravate or initiate BPS/IC symptoms?” with yes, as opposed to 4 patients who denied an association between stress and symptoms worsening. The median answer to the second question “I have a relaxed personality, coping well with stress” was 2.0 (1.0; 2.75) from 0 to 5, which means that patients tend to have a low capacity for coping with stress.

No adverse events were registered, not even in the dropouts.

## 4. Discussion

The most important finding of the present pilot study was the clinical confirmation that the blockade of α1_A_-Ars with silodosin, at a dose of 8 to 16 mg daily, induced a substantial improvement in pain, the primary endpoint. Although the endpoint was defined at 12 weeks, relevant improvement in pain intensity was also observed at week 8. In addition, secondary outcomes such as daily voiding frequency and bladder functional capacity were also substantially improved after 12 weeks of therapy. All these findings certainly contributed to the marked improvement in QoL observed. The effect of silodosin is also in agreement with a previous finding of this group, which consisted of the demonstration of an exaggerated sympathetic activity and high levels of urinary NE excretion [11].

We statistically evaluated the cohort with medians comparisons and with individual treatment responses given the low number of enrolled patients. This way, we weakened the bias that great responses could have in lower responders and at the same time evaluated responses through MID, assuming that a complete resolution of complaints would be unrealistic; rather, a satisfying level of improvement was sought.

All the parameters evaluated improved significantly. This could indicate that silodosin could have a place in the oral pharmacology treatment for BPS/IC patients. These improvements were obtained both in objective and subjective parameters.

In total, 11 out of 17 patients achieved an MID of 3 points of pain in the VAS score, which is a significant result since the main complaint of this condition is bladder pain. These results could be biased by the single-time evaluation of pain, a parameter that could fluctuate over time. When comparing the demographic parameters between responders and non-responders, none of these parameters were found to be helpful in identifying a possible characteristic or predictor of positive results, as seen in Table 2. The urinary frequency, both daytime and night-time, also significantly improved. Although the urge to pass urine in BPC/IC patients is usually caused by pain or the fear of bladder pain, in some patients the high urinary frequency persists despite pain improvement [13], and for this reason the improvement of this parameter could be of equal importance when compared to pain. A total of 14 in 17 patients had an improvement of at least 2 micturitions in total urinary frequency, and 8 of the 17 patients had an MID improvement in nocturia.

The analysis of treatment response *per* patient, using the MID, evaluated the most bothersome features for these patients: pain and urinary frequency. Fifty-nine per cent (10 out of 17) of the patients significantly improved simultaneous in terms of pain and urinary frequency, and this study could strongly support a future randomized clinical controlled trial (RCT), which we are planning to perform. According to the calculations, a total of 70 patients (35 per group) should be randomized to placebo or silodosin in a ratio of 1:1.

The present clinical data support previous experimental observations. Matos et al. showed in rodents that bladder pain, voiding bladder contractions, and histologic bladder, features similar to those seen in BPS/IC patients, could be triggered by the systemic administration of an alpha-adrenergic agonist, and that such effect could be prevented by α1_A_-ARs [14,15]. Moreover, it has also been shown that the activation of peripheral α1_A_-ARs could enhance bladder pain through the pain receptor TRPV1 by increasing the amount of ATP released to bladder distension, suggesting the existence of crosstalk between the sympathetic nervous system and nociceptive pathways.

In addition, in the WAS model of stress, Dias et al. showed increased levels of noradrenaline in the urine of rodents [16]. The WAS model in rodents seems to replicate the signs of nociception and bladder changes seen in BPS/IC patients [11]. Rothrock as well as Lutgendorf have shown that higher levels of stress are related to greater pain and urgency in patients with IC but not in controls [17,18]. Stress threshold differs inter- and intra-individually, as it can be dependent on the circumstances in which the questionnaire is answered. In our study, the answers to both questions seem to reflect a group of patients with a usually low capacity to cope with stress, recognizing symptom aggravation with stressful events. These observations should be considered when selecting BPS/IC patients for alpha-1A blocker treatment and in patient inclusion in RCT.

Silodosin is a very safe drug. The absent effect on α1_B_-AR lowers the risk of dizziness or orthostatic hypotension. In women, we confirmed the safety of silodosin use, even when escalated to 16 mg. Two patients abandoned the study before week 4 at a very early phase of the treatment, eventually indicating that the effect of silodosin on pain is not immediate.

This single-arm study has obvious limitations: the lack of a control group and not being blinded. The number of enrolled patients was small, with three of them dropping out. Intrinsic to this condition, pain is a subjective symptom which depends on different variables, and its intensity can fluctuate over short periods. This last one is specifically true in chronic pain conditions, which can present unprovoked flares. Another limitation was the assessment of self-perception of stress coping, which was based on two questions, and the answer was always modulated by the feeling at that moment.

## 5. Conclusions

In conclusion, daily silodosin could be an effective and well-tolerated treatment for refractory BPS/IC females. The treatment effect was sustained for 12 weeks. Personality type does not seem to be an independent factor for treatment response. More robust studies should be conducted to assess the role of silodosin in BPS/IC patients.

## Figures and Tables

**Figure 1 jcm-11-05659-f001:**
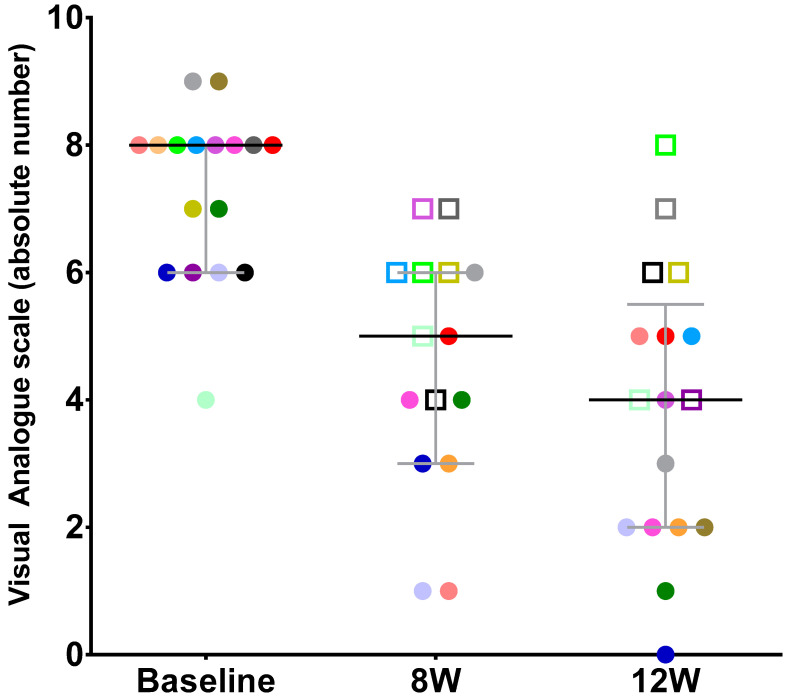
Median and (p25–p75) intervals and individual pain score on the VAS at baseline and weeks 8 and 12. Each patient is identified by a specific color and squares mean that MID was not reached.

**Figure 2 jcm-11-05659-f002:**
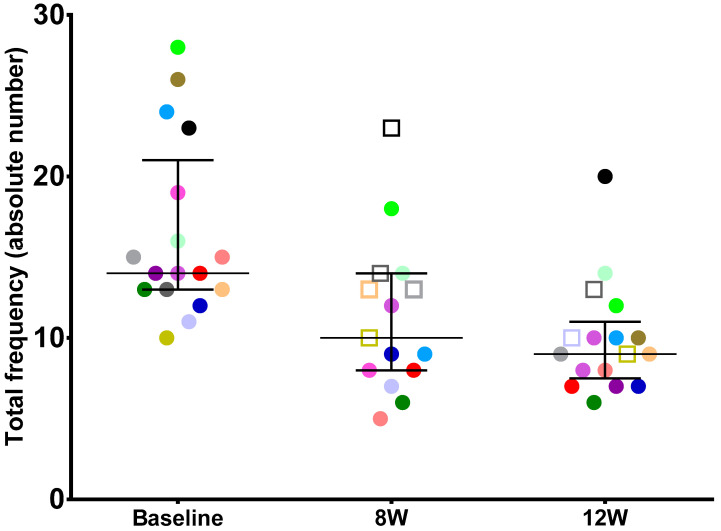
Median and (p25–p75) intervals and individual total daily frequency at baseline and weeks 8 and 12. Each patient is identified by a specific color, and, when represented as a square at 8w or 12w, this means that MID was not reached.

**Figure 3 jcm-11-05659-f003:**
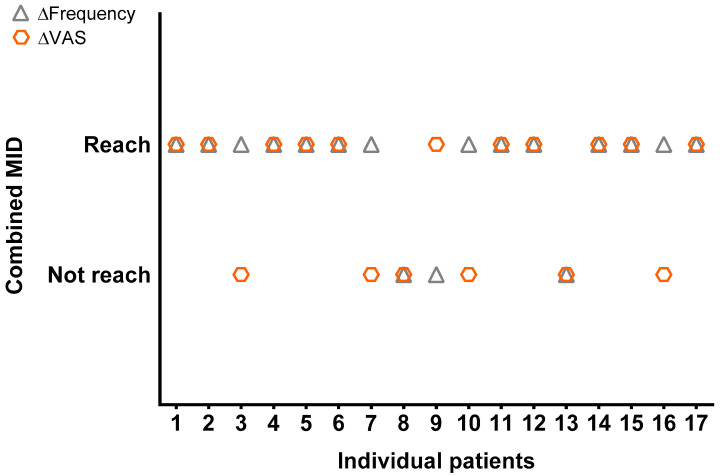
Individual combined evaluation of MID in VAS score and total urinary frequency at week 12.

**Table 1 jcm-11-05659-t001:** Outcome comparison between baseline and 12w.

	Baseline Median (P25, P75)	W12 Median (P25, P75)	*p*-Value
Pain VAS	8.00 (6.00; 8.00)	4.00 (2.00; 5.50)	0.001
Total Voiding Frequency	14.00 (13.00; 21.00)	9.00 (7.50; 11.00)	0.001
Nocturia	4.00 (2.00; 5.50)	2.00 (1.00; 3.00)	0.001
MVV	175.00 (150.00; 250.00)	250.00 (200.00; 350.00)	0.03
OSS	26.00 (24.00; 28.00)	19.00 (14.00; 22.50)	0.001
QoL	5.00 (4.25; 6.00)	3.00 (2.00; 4.00)	0.001

**Table 2 jcm-11-05659-t002:** Comparison of demographic characteristics between patients that reached the primary endpoint (responders) and those who did not (non-responders).

Demographics	Responders	Non-Responders	*p-*Value
Age	49.0 (35.0; 67.0)	58.8 (49.0; 64.0)	0.52
Baseline VAS	8.00 (7,00; 8.00)	6.50 (5.50; 8.00)	0.09
Total Urinary Frequency	14.00 (13.00; 19.00)	15.00 (12.50; 24.50)	0.81
Nocturia	4.00 (3.00; 5.00)	3.00 (1.75; 8.00)	0.81
MVV	150.00 (150.00; 250.00)	200.00 (150.00; 265.00)	0.78
OSS	24.00 (27.00; 29.00)	25.00 (23.50; 27.50)	0.24
QoL	5.00 (4.00; 6.00)	4.50 (6.00; 3.80)	0.40

## Data Availability

The data presented in this study are available on request from the corresponding author. The data are not publicly available due to privacy issues.

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
