# Peer review of "Silodosin Improves Pain and Urinary Frequency in Bladder Pain Syndrome/Interstitial Cystitis Patients"

_jcm, 2022, doi:10.3390/jcm11195659_

Round 1

Reviewer 1 Report

The article deals with the efficacy of silodosin to be used as an off-label drug in Bladder pain syndrome / interstitial cystitis BPS / IC.

The authors have produced an exciting and quality article.

The authors start from the hypothesis that there is a correlation between circulating norepinephrine levels and the specific clinical manifestations of BPS / IC, as high levels of norepinephrine would stimulate the bladder alpha-adrenergic receptors. Various studies in the literature support this hypothesis, which authors report in their article both during the introduction and discussion. Starting from this hypothesis, the authors proposed an adequate study design, selecting patients with BPS / IC to administer a selective bladder alpha-adrenergic receptor inhibitor, silodosin. The drug, tablets from 4 mg to 8 mg, was administered to the patients included in the study. The patients are then checked at zero time, at 8 weeks and at 12 weeks to evaluate the effects of the therapy through an outpatient visit in which the effectiveness of the therapy is investigated with adequate pain assessment scales. The selection criteria of the Patients are adequate; no particular BIAS selection is detected; the exclusion criteria are also adequate, as they exclude the Patients subjected to systemic or local painkilling treatments. The effect of silodosin is thus evaluated.

The statistical sample is small but remains adequate for the study as a pilot study. The study's objectives are divided into primary and secondary; the primary endpoint is the statistically significant reduction of pain with the administration of silodosin. The primary endpoint is reached, and the static analysis is adequate for the type of study and considering the small number of patients enrolled. Secondary endpoints are clinical improvement of various symptoms associated with BPS / IC. Secondary endpoints are met. The study has a meaningful clinical impact because it introduces the possibility of using a drug with reduced side effects for a very invalidated pathology for the patients affected. The study encourages us to continue on the path of this analysis with further studies, increasing the number of associated patients. We only suggest dividing the patients into two arms and administering a placebo to one arm, as BPS / IC is strongly associated with disorders from somatization stress. Finally, we do not detect any violations of the journal's editorial rules.

Author Response

Dear colleague,

Thank you for your kind words. This was the pilot study, and the results obtained will be used to calculate the power and sample size of the randomized controlled trial that will be used (we added this information to the paper in the Discussion section), but a mean reduction of 4.0 points in the VAS pain score was reported in this study. Assuming a placebo effect of 50% of the silodosin benefit, a mean pain VAS reduction of 2 points after 3 months with placebo was considered. Thus, the minimum difference expected between silodosin and placebo was settled at 2.0 points. Assuming a power of 80% and a two-sided significance level of 5%, the calculation determines that 70 patients could be required to complete the study (35 patients per group), already considering dropouts.

Thank you for the kind comments, once again.

Reviewer 2 Report

The authors describe a study investigating the effect of silodosin on pain and urinary frequency in BPS/IC female patients.

Title:  The title is clear, concise, and accurate.  

Abstract:  The abstract is well written and definitions of the population are sufficiently well stated in the methods.

Materials and methods: Methods are overall well described and thoroughly explained. However, it would be great to have some details on results of uroflowmetry in your subjects. I think it can be also useful to have the analysis of post-void residual before and at FU visits.

Results, Discussion- these sections are presented in a well- organized manner.

I have one major comment to results section- from clinical perspective it seems to very unusual complete lack of any AE’s. Does it really happen in 3 months of observation (i.e. no headache, backpain or bladder pain at all?)? I think the authors should more carefully ask about AE’s in future studies.

What was the reason of drop-outs? I would like to encourage to authors to add more details describing that sub-group in revised paper.

Author Response

Dear colleague,

Thank you for your words.

Unfortunately, the uroflowmetry was not performed before or after. However, patients had a 0cc of post-voiding residual urine and none complained of straining/long time to empty the bladder which leads us to assume an “efficient” micturition. The reasons of drop out are now better explained in the Results section  - demographics.

Thank you for the kind comments, once again.